# The Food Environment of Primary School Learners in a Low-to-Middle-Income Area in Cape Town, South Africa

**DOI:** 10.3390/nu13062043

**Published:** 2021-06-15

**Authors:** Siobhan A. O’Halloran, Gabriel Eksteen, Nadene Polayya, Megan Ropertz, Marjanne Senekal

**Affiliations:** 1Faculty of Medicine, Institute of Medical Sciences, University of Oslo, 0372 Oslo, Norway; siobhan_au@hotmail.com; 2Division of Physiological Sciences, Faculty of Health Sciences, University of Cape Town, Cape Town 7935, South Africa; abrieeksteen@gmail.com (G.E.); nadene1008@gmail.com (N.P.); mnropertz@gmail.com (M.R.)

**Keywords:** food environments, school, home, community, food, diet, obesity, overweight, children

## Abstract

Rapid changes in food environments, where less nutritious foods have become cheaper and more accessible, have led to the double burden of malnutrition (DBM). The role food environments have played in shaping the DBM has attained global interest. There is a paucity of food environment research in low-to-middle-income countries. We conducted a case study of the food environments of school aged learners. A primary school in Cape Town was recruited. A multi-method design was used: a home food and eating behaviours questionnaire completed by 102 household respondents and four questions completed by 152 learners; learner participatory photography; a semi-structured school principal interview; a tuckshop inventory; observation of three-day tuckshop purchases. Foods that were commonly present in households: refined carbohydrates, fats/oils, chicken, processed meats, vegetables, fruit, legumes, snacks/drinks. Two thirds of households had rules about unhealthy drinks/snacks, ate supper together and in front of the TV, ate a home cooked meal five–seven times/week and ate breakfast together under two times/week. Vegetables were eaten under two times/week in 45% of households. A majority of learners (84%) took a lunchbox to school. Twenty-five learners photographed their food environment and 15 participated in semi-structured interviews. Six themes emerged: where to buy; what is available in the home; meal composition; family dynamics; peer engagement; food preparation. Items bought at informal food outlets included snacks, drinks and grocery staples. The principal interview revealed the establishment of a healthy school food environment, including a vegetable garden, although unhealthy snacks were sold at the tuckshop. Key dimensions of the food environment that require further investigation in disadvantaged urban and informal settlement areas include the home availability of unhealthy foods, eating behaviours in households and healthfulness of foods sold by informal food outlets.

## 1. Introduction

The United Nations (UN) declared 2016–2025 the Decade of Action on Nutrition and aims to increase government policies and action to end all forms of malnutrition, which includes undernutrition, micronutrient deficiencies and overweightness and obesity [1]. Unhealthy diets are an increasingly important risk factor for obesity and diet-related non-communicable diseases (NCDs) globally [2].In South Africa, adolescents, especially urban females, were burdened by overweightness and obesity with prevalence increases of 6% in boys and 7% in girls between 2002–2008 [3].

A key factor contributing to unhealthy diets is the food environment [4]. The food environment is defined as the ‘collective physical, economic, policy and sociocultural surroundings, opportunities and conditions that influence people’s food and beverage choices and nutritional status’ [4,5,6,7]. Changes in agricultural systems and technological advancement have led to a global shift away from healthy foods to food environments dominated by heavily promoted, widely available, cheap and convenient, energy dense and nutrient poor foods (EDNP) [8,9].The role that food environments play in shaping population dietary intake has gained notice on a policy level [10,11], set against the backdrop of the United Nations Sustainable Development Goal (SDG) 2 to end hunger, achieve food and nutrition security, improve nutrition and promote sustainable agriculture [11]. Global actions to improve population nutrition and reduce the risk of diet-related NCDs, to achieve the SDG, involve increasing comprehensive monitoring efforts and indicators related to nutrition polices and food environments [12]. An example is the nine-module framework developed by the International Network for Food and Obesity non-communicable diseases Research, Monitoring and Action Support (INFORMAS) [5]. Thirty countries have implemented this framework to monitor, benchmark and support actions to create healthy food environments and reduce obesity risk and non-communicable diseases (NCDs) [5,13].

Food environment research has been primarily undertaken in high-income countries (HIC) [14,15,16]. Although there is a growing body of literature emerging in low-and-middle income countries (LMICs), research in the field is in the early stages [17]. In South Africa, most studies have examined either the home, community or school food environments. For example, three studies have explored the availability and purchasing of foods in schools and shown that there is scope for improvement towards access to healthier food options [18,19,20]. Evidence from the community has shown that small, market-based vendors sell unhealthy foods in cities and rural areas [21] and there is an unequal distribution of supermarkets across high-and low-income urban areas [22]. Data at the household level highlight changes in adolescent dietary habits [23] and the targeted television marketing and promotion of unhealthy foods to children [24,25]. To advance the field of food environment research in the South African context, we aimed to conduct a case study to compile a comprehensive profile of the home, community and school food environment of primary school aged learners. 

## 2. Materials and Methods

### 2.1. Study Design

A multi-method design was used that involved qualitative methods (a key informant interview with the school principal and a Photovoice component with the learners) and quantitative methods (school tuckshop purchases observation, a household respondent questionnaire and a learner questionnaire). The community food environment was conceptualised as the type, location and accessibility of food outlets [26] and peer influences were also explored. The home food environment was defined as food availability and related behaviours in households [27]. The school food environment was examined in terms of what foods are provided (e.g., school lunch program) and sold (e.g., foods available in the tuckshop), promotion of healthy eating and school lunchbox practices [28].

### 2.2. Study Area and School Recruitment

We aimed to recruit a school in the Cape Town metropole in South Africa that met the following criteria: an operational school tuckshop; participation in the National School Nutrition Programme (NSNP); willingness of the principal to be interviewed. In addition, the area around the school needed to be safe for the researchers and for formal and informal food outlets to be present within a 400m radius of the school. The NSNP is a government programme which provides a nutritious meal (usually consisting of a starch and protein dish with vegetables), which is prepared on the school premises and is provided to learners as required. The programme also promotes healthy eating behaviours in learners via an education course and encourages the development of school gardens [29]. The selected radius was based on previous research which demonstrated an association between the proximity of food outlets within a 400 m radius of schools and improved health outcomes [30]. In South Africa, formal food outlets include neighbourhood convenience stores, specialty stores, chain supermarkets and large wholesale and retail stores [31] Informal food outlets include General Dealers (non-registered retail stores), ‘Spaza’ shops (the most common food outlet within the informal sector, often located in townships and poorer neighbourhoods), street vendors (cheap, often lower quality foods sold from roadside stalls, may be permanent or highly mobile) and home shop vendors (inexpensive, poor quality foods sold from a small shop in the front of a home) [31]. 

The recruited school was a quintile 4, mixed gender, medium-sized, public primary school with English language as the teaching medium, located in a northern suburb in the urban city area of Cape Town. Historically, this part of the city was designated as a coloured area during the apartheid era (1948–early 1990s) [32]. Public schools across South Africa are grouped into five quintiles according to the demographics of the neighbourhood, where schools in quintile 1 are the poorest and those in quintile 5 the least poor [33]. Although classification as a quintile 4 or 5 school is based on national demographic data, enrolments are not always drawn from the local area. Learners from families living in poorer, distant areas may still attend a quintile 4 or 5 school [33]. An example is our recruited school, where many of the learners were transported each day from 18 different formal and informal (areas with improvised buildings and lacking adequate infrastructure) settlement areas, some located 20 km away from the school. Often, a school bus or mini-bus taxi was the only mode of transport available to learners from these areas. The total number of learners across all grades (grades one to seven) in the school at the time of the study was 668 (typically aged 6 to 13 years), with an approximate equal distribution of boys and girls.

### 2.3. School Principal Interview

The principal key informant interview took place at the school. It was conducted by a trained dietitian (G.E.), using an audio recorder and an interview guide (included as Appendix A, Appendix A) and took 90 min to complete. The interview guide consisted of 16 semi-structured questions adapted from a manual used in similar unpublished research. The interview guide questions were reviewed and refined for this research by a panel that consisted of three senior academics and four post-graduate students. In the finalised version, the included questions explored the principal’s views on the learner profile and general health (e.g., health screening of learners), health promotion education and activities at the school, experience of the NSNP, healthfulness of foods/snacks, the school tuckshop and the influence of teachers on learner’s food choices.

#### Interview Analysis

The interview was transcribed verbatim for analysis (S.A.O.) and checked for consistency and local language interpretation (G.E.). The transcription was read independently (M.S. and S.A.O.) for familiarisation with the content. The interview data was thematically analysed interactively by M.S. and S.A.O., where themes were identified and organised via an inductive process without analytical preconceptions [34]. Differences in themes identified between the two researchers were discussed and a consensus agreed. The thematic framework that emerged from the interview is included in the Appendix A Appendix A. Verbatim quotes that illustrate themes/subthemes were not retrieved as the principal consented only to the publication of an approved summary of his views.

### 2.4. Tuckshop Observation

Two fieldworkers conducted an inventory of all items stocked in the tuckshop, which was used to develop a checklist for the observation of purchases made by learners in grades four–seven over a three-day period during first and second break times. Transaction details were manually recorded on the checklist when a transaction was performed. One fieldworker recorded purchases made by girls and those made by boys were recorded by the other. Details recorded included: date, gender, break time (first or second break), the type, number and cost of items purchased.

#### Data Analysis 

Analysis included tallying frequencies for categorical variables and calculation of the median (inter quartile range) (IQR) for continuous variables, which were all non-normally distributed (Schapiro Wilk test *p*-values of <0.05) using Statistica version 13.2 (Tibco, Palo Alto, CA, USA). Comparisons between genders was performed using the Mann–Whitney U test.

### 2.5. Photovoice 

#### 2.5.1. Background

The Photovoice methodology [35] was used to elicit an understanding of the varied food environments of grade seven learners. Photovoice is a qualitative tool used for participatory research [35,36] and uses the immediacy of the visual image to allow participants to record their diverse surroundings [36]. Interviews with participants about their photos provide definition of the images and control over the selection of photos to best represent their experience [36]. Photovoice has been shown to be an effective approach for gathering adolescent input in community change [37,38,39] and to gain children’s perspectives on their environments regarding physical activity and eating behaviours [40,41,42,43]. 

#### 2.5.2. Participant Recruitment 

The target population were grade seven learners and the target sample size was 35, based on previous research [40]. All learners present on the day of recruitment in September 2019 were eligible to participate. Learners received a short information session from fieldworkers and received parent/primary caregiver consent forms to take home. Learners who returned signed consent forms and who also consented to participate were included in the photovoice component of the study. 

#### 2.5.3. Procedures and Interviews

Each participating learner (*n*=35) received a disposable camera for a one-week period and were provided with training from the fieldworker on camera usage. This included guidance on the type of photographs required to capture their food environment, e.g., fridge and kitchen cupboard contents, meals, friends or family members who shared their meal, food outlets and any other food related photographs they believed illustrated their food environment. Learners were provided with safety advice (e.g., avoiding photographing in areas where they felt unsafe) and were trained on obtaining permission from the shop owner/employee/individual, prior to taking the photograph. 

A total of 25 cameras were returned. Reasons for drop-out included damage to the camera (*n* = 1), late camera return (*n* = 5) or absence on interview day (*n* = 4). Photographs were examined by M.S. and N.P. and a maximum of five photos that reflected each participant’s home and community food environment were selected for discussion in an individual interview. Semi-structured interviews were conducted with learners at a school venue and audio recorded by N.P. within two weeks after the return of the cameras. The semi-structured interviews were an adapted version of the SHOWED methodology [44]. The SHOWED method involves structuring Photovoice interviews by posing a series of questions about the participant’s photos (in this study, three–five questions), and were framed to elicit a descriptive response, e.g., “What kind of meal is this?” “How often do you eat these types of foods?” “Where do you or your family buy this?” Additionally, and guided by N.P., learners discussed how their photos related to their diet and food availability and accessibility in their environment [44]. A total of 15 interviews were conducted by N.P. and took place in the last week before the school break. The reason for drop-out was absence from school on the designated interview day. Each learner received a small gift of stationary upon return of the cameras and a healthy snack at the conclusion of the interviews.

#### 2.5.4. Interview Analysis 

Interviews were transcribed verbatim by N.P. Each interview was read independently by M.S. and N.P. to identify common themes across the interviews. A coding list was developed inductively by hand in interactive sessions between M.S. and N.P. Each theme was assigned a four-digit code: the first, second, third and fourth digits represented the theme, subtheme, sub-subtheme and the sub-sub-subtheme, respectively [34]. For each interview, any reference to a theme was coded accordingly and the frequency of mention for each code was recorded in an Excel spreadsheet (included as Appendix A Appendix A). For quality control purposes, M.S. and a third researcher (S.A.O.) (who read the interviews independently), then checked the code allocation interactively. Any discrepancies were discussed, and a consensus agreed on between M.S. and S.A.O. Themes and subthemes are mentioned specifically in the results, with sub-sub and sub-sub-subthemes reflected in examples given and text quotes.

### 2.6. Household Respondent Questionnaire and Learner Questions

#### 2.6.1. Participant Recruitment

All grade five–seven learners (aged 10 to 13 years old) present in the school at the time of the study (*n* = 228) received a 15 min information session about the research in September 2019. During this session, all learners in each of the three grades, including those learners who participated in the Photovoice study component, were invited to participate in the quantitative part of the research, which included completing the questionnaire. Four questions from this questionnaire which were most relevant to this study are reported on in this paper. Learners were also invited to engage a representative from their households (e.g., a mother, father, grandmother, aunt, sister or brother) to complete a household questionnaire. Each learner received the following to take home: a consent form for him/her, the learner questionnaire to be signed by a parent/primary caregiver and a household food environment questionnaire to be completed by a household representative. The latter also included an information sheet and consent form to be signed by the household respondent. Completed household questionnaires were collected by classroom teachers and were returned to the fieldworkers. Learners who returned completed consent forms for participation in the learner questions also provided written consent. 

#### 2.6.2. Learner Questions 

The questions completed by the learners were: (1) one closed-ended question which queried if lunchboxes were taken to school; two open-ended questions that probed lunch box preparation and contents and (2) one free text question probing the food items/dishes and snacks/drinks their best friend liked and disliked the most. All questions were interviewer administered. 

#### 2.6.3. Household Questionnaire

The questionnaire was composed of two sections which included a total of 18 questions. The first section consisted of nine sociodemographic questions which were derived from a previous study [45]. Food security was assessed by the Community Childhood Hunger Identification Project (CCHIP) index which is composed of questions regarding household-level food insecurity (three items), individual-level food insecurity (one item) and child hunger (four items) [46]. A score of 5 or more was deemed to reflect hunger, a score of 2–4, risk of hunger and a score of 1, no hunger [46]. 

The second section included nine household food and eating questions which were derived from questionnaires previously used in similar research in Cape Town (unpublished), the United Kingdom [47] and the United States [48]. Four closed-ended questions covered rules about the types of foods that their child may eat; factors that may influence what their child eats; their likes and dislikes of fruits and vegetables; their encouragement of their child around eating food and meals. One question explored if meals were eaten together as a family and/or whilst watching TV. Response options for these two questions included a 5-point Likert scale ranging from not every week/never to one or more times/day. One question probed where the household respondent did their food shopping. Two free text questions probed which food items/dishes and snacks/drinks household respondents liked and disliked the most. The availability of household food items was determined via a household inventory adapted from a previous study [49]. In instances where the household respondent was illiterate, learners were instructed to assist with the completion of the self-administered questionnaire.

#### 2.6.4. Data Analyses

Statistica version 13.2 was used for analysis of the household and learner data. Frequencies were tallied for categorical variables and the mean (standard deviation) (SD) or median (inter quartile range) (IQR) for continuous variables was calculated depending on the distribution of the data according to the Schapiro Wilk test. Responses to the free text questions were categorized and collapsed for reporting. This resulted in 13 categories for food items/dishes most liked and least liked. These categories included: pasta dishes; starches (maize porridge, rice and cereals); vegetables; fruit; chicken; red meat; fish; meat-starch dish; energy dense dishes/foods (fried chicken, burgers, pizza and meat pies); stews; bread and topping; other (chocolate, pudding, cake, jam, crisps, dairy, legumes; organ meat, eggs; soup; proportion mentioned <4% per item); do not know. The six categories for favourite snack/drink included: sugar sweetened beverages (SSBs) (mentioned with or without an unhealthy snack); salty snacks (crisps and biscuits if not mentioned with a SSB); sweet snacks (sweets, chocolate, cake, biscuits if not mentioned with a SSB); healthy snacks (yoghurt, fruit and popcorn); other (chicken, take-outs, pies, bread, fruit juice, tea/coffee; proportion mentioned <4% per item); do not know. 

## 3. Results

### 3.1. School Principal Interview

The three environmental settings, namely school, community and home environmental settings that emerged as themes, and subtheme results, are presented accordingly. 

#### 3.1.1. School Food Environment

The principal described the educational component of the NSNP, where learners are taught about food and nutrition (life skills program). His opinion was that the program makes an important contribution to the learners’ understanding of healthy eating. The principal also described the type of school meals provided two years ago and the changes that had since been implemented. He indicated that a sugary, flavoured milk drink that the children liked because of the sweetness was provided in the previous NSNP. Currently the NSNP includes a basic meal containing a protein, e.g., soya, and a starch. The principal emphasised that the inclusion of fresh vegetables and fruit in the meals was an important dietary component for the learners.

The principal also talked of the establishment of the school vegetable garden, which included growing a variety of vegetables hydroponically and in raised garden beds. He pointed out that learners were actively involved in the garden upkeep (Figure 1). He also revealed that the grown vegetables (e.g., spinach) are added to the NSNP meals, which he believed improved learners’ daily vegetable intake.

The principal expressed a desire to limit the type of unhealthy snacks and drinks sold from the school tuckshop and to increase the provision of healthy options. However, he mentioned that the learners often did not have enough money to buy healthy foods. He also spoke of the financial importance of the tuckshop and the income it provided, which enabled the purchase of stationary and to maintain the school building and premises. The principal emphasised the importance of positive teacher role modelling. He explained that some teachers consumed SSBs during school hours, but he felt that it was not within his remit to demand that they do not do so. He personally preferred to model healthy eating behaviours and encouraged this approach amongst the teachers at his school.

#### 3.1.2. Community Food Environment 

The principal described the learner’s neighbourhoods as low resourced, poor and unsafe with gangs intimidating local communities and committing gun crime, which may have impacted parent’s access to local shops and their ability to buy food for their family. 

According to the principal, food outlets in the community consisted of informal street vendors and informal home shops which learners accessed on their way to school. The learners often bought multiple packets of chips from these vendors and took them to school. He voiced concern about the poor quality of the food and unhealthy snacks sold by the street vendors. When asked about the feasibility of restricting the types of foods street vendors sell, he suggested that one possibility would be to establish a school policy that addresses this issue but admitted that there would be a number of challenges in implementing such a policy.

There had been some effort by the principal to improve the healthiness of the items sold from a street vendor located close to the school, where fruit was made available for sale for a number of months. On occasion, some learners bought pieces of fruit, but most did not and subsequently the street vendor, who was a parent of one of the learners at the school, stopped providing fruit for purchase. The principal had also attempted to prevent learners from buying certain snacks from other street vendors, located outside the school premises. However, he had to concede that one vendor, who travelled some distance to be there at 5:30am, did provide a service to learners who arrived at school early and that prohibiting the sale of these items might affect the vendor’s livelihood.

#### 3.1.3. Home Food Environment 

It emerged from the interview that many of the learners originated from lower socio-economic areas, characterised by informal housing. It was the perception of the principal that insufficient food at home for the family and busy working mothers, who may not have the time to prepare a school lunch for their child, meant children were given a small amount of money to purchase something to eat on their way to school instead. In his experience, some learners often arrived at school with biscuits and SSBs; however, he emphasised that the reason was not because the parents were unconcerned about their child’s eating habits, rather that they were struggling to make things work in their daily lives.

Regarding the family structure, it emerged that learners’ families comprised of a mother and a father, or a single parent. It was the view of the principal that working parents often encountered issues with their employers, which may negatively impact their capacity to care for the child/children and contributed to their struggles in life. It further emerged that it was his view that parent drug use and/or alcohol consumption may have affected child caring capacity and that in these situations a grandparent was responsible for looking after the child/children.

### 3.2. Tuckshop Observation 

Five categories of snack foods were available for purchase at the school tuckshop, with the sweets category having the greatest number of product types (derived from the tuckshop inventory) (Figure 2).

In total, 290 transactions took place over the three-day period. Girls made 54% of these transactions and boys 46%. The median (IQR) number of items purchased per transaction was 3 (2; 4) and the amount of money spent per transaction was 2 Rand (equivalent to ~10 p GBP) (2; 4.5). These two variables did not differ significantly between boys and girls; median (IQR) items purchased: 3 (2; 5) and 2 (2; 4) for boys and girls respectively and amount spent per transaction: 2.5 (2; 5) and 2 (1; 4.5) for boys and girls respectively). Boys purchased a greater variety of items than girls: median (IQR) of 2 (1–2) vs. 1 (1–2), Mann–Whitney U-test *p* = 0.025. The majority of transactions were made during first break (65%), however the median (IQR) number of items purchased per transaction was similar during first [3 (2; 4)] and second breaks [3 (2; 5)].

### 3.3. Photovoice Results

All learner-contributed photos were considered in the data analysis, although the number of photos taken by each learner varied. Approximately three–five of the photographs that depicted the food environments were selected for the interviews. Learners took pictures predominantly of their meals and the foods available in the home, and to a lesser extent, foods available in their community and school environment. Six themes were identified, presented across the interviews and food environment related photographs: (1) where to buy; (2) what is available in the home; (3) meal composition; (4) family; (5) peer engagement; (6) food preparation.

Theme 1: ‘Where to buy’. Learners described a number of places where they acquired foods and beverages and six sub-themes under ‘where to buy’ emerged; (1) informal home shop vendors; (2) supermarkets; (3) street vendor; (4) formal food outlet; (5) mother’s work; (6) tuckshop (Figure 3). 

The most commonly mentioned place where learners and their household members purchased food items was ‘informal home shop vendors’, located close to the learner’s home. Purchases from these outlets were mentioned to be made daily, with ‘snacks and fizzy drinks’ being commonly photographed and mentioned items, followed by grocery items (e.g., maize meal, spices, bread).When learners were asked about how often they visited the vendor, a number of them replied, ‘*every-day*’. Supermarkets also emerged as a place where foods and drinks were purchased. Learners and their household members bought ‘groceries’ from these shops, which included items, such as ‘*the normal type of cheese that we buy from the shop like Britos or Pick ‘n Pay*’.

Foods and drinks were also bought from formal food outlets (e.g., fast food outlets) and many of these outlets were located close to the learners’ homes, although learners mentioned that purchases were not made daily. Some learners reported their ‘mother’s work’ as a place to buy foods, especially snacks. When a learner was asked about where a packet of popcorn was purchased, he replied, ‘*My mom bought it at work*’. Conversely, street vendors were only mentioned or photographed by a few learners as places where they bought food and beverage items. One learner pointed out ‘*This, like, um, people always by the road sell them so when I pass and I see chicken feet, then I buy*’.

Although there were no photographs taken of the school tuckshop, a few learners mentioned the tuckshop as a place of purchase, when questioned about the photos of items consumed at school. Foods acquired from the school tuckshop included snacks, such as ‘*chippa (corn-based savoury snack) and snickers and lays (crisps) and stuff*’.

Theme 2: ‘What is available in the home’. A wide variety of foods available in the home were represented in the photographs and eight subthemes were identified: (1) protein; (2) starch; (3) vegetables; (4) fruit; (5) fats; (6) snacks; (7) beverages; (8) dairy. A number of learners explicitly mentioned fresh vegetables and meats such as ‘tomatoes’, ‘potatoes’, ‘red meat’ and ‘chicken’. To a lesser extent, more refined foods such as ‘white bread’ and ‘maize meal’ were mentioned, while ‘mayonnaise’ and ‘oils’ were commonly mentioned as fats present in the home. Energy dense, nutrient poor foods (EDNP) such as ‘chips’, ‘sweets’, ‘chocolates’, ‘cakes and muffins’ and ‘fizzy drinks’ (SSBs), although photographed, were not often talked about. The majority of the learners took photographs of food items available inside the home (Figure 4 and Figure 5). One learner photographed her vegetable garden where the family grew a vegetable called “covo” (kale), used in stews and sauces.

Theme 3: ‘Meal composition’. The five sub-themes under meal composition were: (1) protein, vegetables and starch; (2) protein and mostly starch; (3) only starch; (4) bread meals; (5) special meals. Of the five sub-themes, ‘protein, vegetables and starch’ emerged as the most common meal composition and learners often mentioned ‘red meat’, ‘onion and tomato’ and ‘maize meal’ when describing what foods made up their meal. Other meal components included ‘salad’, ‘chicken’, ‘rice’ and ‘potatoes’. When asked if potatoes were usually a large part of meals, the learner responded ‘*Yes… [In]*
*Almost everything we eat. Every day we eat potatoes’* (Figure 5).

Bread meals were defined as meals that included bread as the key item, e.g., sandwiches, that were either taken in lunchboxes or eaten at any time during the day (Figure 6). White bread was the most commonly mentioned type of bread eaten, with polony (highly processed luncheon meat made from pork, beef or chicken) as the most frequently mentioned filling. When a learner was asked why they eat white bread they replied, ‘*I like it because it doesn’t make me full and I don’t want to be full in school*’. When asked why they did not like to be full, they replied, ‘*Because I might get tired or I might sleep*’.

Children considered ‘Sunday meals’ as ‘special meals’ and eating these types of meals together (e.g., braai/barbeque meats, certain types of vegetables and fruits) was often talked about by a number of children during the interviews. One child stated, ‘*Sundays we eat with some vegetables and Saturdays we eat differently in the afternoon*’. When asked how different a Sunday plate would look, another child replied ‘*it would look like more food, more colour, different kinds of food*’.

Theme 4: ‘Family’. The five subthemes under family were: (1) eating together; (2) where they eat; (3) what is eaten together; (4) family likes; (5) family dislikes. Learners often reported not eating together and eating ‘in front of the TV’ and ‘at the table’, and a few learners mentioned that the TV was placed in the bedroom, and this was then described as the place to eat a meal (Figure 7). Snacks such as ‘chips and sweets’ seemed to be typical items that learners ate with their families, with peanuts, slangetjies (a jelly based sweet) and mageu (a fermented maize drink) being the most commonly mentioned. One learner said [I eat] ‘*peanuts, raisins and different color dried fruits, with my mother and my sister*’. Other foods that families seemed to like included vegetables, meals and snacks. One child stated ‘*My mommy likes to have vegetables*’ and another said ‘*I’m not a lover of polony, but my mommy and daddy is*’.

Theme 5: ‘Peer engagement’. The four subthemes under peer engagement were (1) eating with peers; (2) what they eat together; (3) where they eat; (4) likes. Learners mentioned consuming ‘snacks’ (e.g., ‘sweets’, ‘chips’) with their friends ‘at school’ more often than at ‘each other’s houses’. When the learner was asked about whom she likes to eat these snacks with, she replied, ‘*With my friends*’. When asked where these snacks were eaten, the learner replied, ‘*It was in the morning, here at school’* (Figure 6).

Theme 6: ‘Food preparation’. The three subthemes under food preparation were: (1) who prepares meals; (2) equipment; (3) preparation methods. Learners identified their ‘mother’ as the person in their household who prepared the meal most often. For example, when asked about a photographed pasta dish, the learner responded, ‘*my mummy makes it, but she makes the tomato one*’.

Learners cited the use of a two-plate stove, gas stove and a paraffin heater as equipment used to prepare food. As one learner said, ‘*If my mother doesn’t want to cook on the stove, if she’s lazy to stand up, then she lights on the heater and puts the pot on the heater and the food cooks on the heater*’. When asked about the cooking methods used to prepare their meals, learners mentioned ‘stews’ and ‘frying’; for example, a learner said ‘*When we cook rice with stew then we put potatoes and with mielie* [corn on the cob]’. Another learner said ‘*It’s the chips you fry to eat.’* When asked how a particular learner’s household eats their vegetables, the reply was: ‘*Mixed together with meat. In a stew*’ (Figure 7). 

### 3.4. Household Profile Results

#### 3.4.1. Socio-Demographics 

The household questionnaire was completed by 101 respondents, who were most likely to be a parent of the learner (92.8%). The median age of the total group of household respondents was 39 years, with two-thirds of the household respondents being married. One third of household respondents had some high school education. 

Households comprised a median number of two adults and two children. One in five households experienced hunger, compared to half of the households which experienced no hunger (Table 1).

#### 3.4.2. Household Eating Behaviour

Results on household eating behaviour are presented in Table 2. Supper was regularly (five–seven times/week) eaten together by family members in 65.7% of households, while breakfast was eaten together by family members in 25.7% of households. Regular eating of a meal while watching TV was reported for 59% of households, while snacks were consumed regularly while watching TV in 42.4% of households. A home cooked meal was eaten regularly in 72.2% of households and vegetables were regularly eaten in 26% of households (Table 2). 

Almost three-quarters of household respondents agreed that they eat the food they want their children to eat and agreed that they encouraged their children to eat vegetables, fruit, brown/whole wheat bread, all the food on their plates and to sit at the table when eating meals. Half reported that cultural factors determined which foods they eat as a family (Table 2). The majority of respondents agreed they like the taste of most fruit, liked most vegetables, liked tasting new fruit and liked tasting new vegetables (Table 2). The majority of households had rules in the home relating to the consumption of fizzy drinks; fat cakes, doughnuts and slap chips (chips fried twice); sweets and chocolates; sweet biscuits, tarts and cakes; take-out foods; crisps; pies, samosas and sausage rolls; sugar (Table 2). 

More than 60% of respondents agreed that children’s knowledge of healthy eating; whether children take a lunch box to school; what children see for sale at tuck shops, vendors and shops; their body image influence what they eat (Table 2). Half or fewer of the respondents agreed that what other people living with a child eat and drink; what children see in advertisements, on TV and on billboards; what children’s friends eat and drink; children’s parents’ working hours; what children’s teachers eat and drink influence what they eat (Table 2).

#### 3.4.3. Household Food Purchasing 

Results on where households purchased foods are also presented in Table 2. Foods were purchased at Spaza shops infrequently (twice or less a week) by 66.3% and regularly (five–seven times a week) by 21.5%; at Cafés infrequently by 81.9% and regularly by 9.6%; at General Dealers (non-registered retail stores) infrequently by 60.2% and regularly by 28.5%; at chain supermarkets infrequently by 56% and regularly by 25.2%; at wholesalers infrequently by 84% and regularly by 9.2% of households (Table 2). 

#### 3.4.4. Household Food Inventory

The top four foods that were available in the majority of households (>90%) were starches (samp, pasta), oil, sugar and fats (Table 3). Other foods that were available in 80–90% of homes were onions, dairy, chicken, potatoes, eggs and tomatoes, whilst peanut butter and fruit, white bread, legumes and maize meal were available in 70–80% of homes. Carrots, viennas, jam, cheese, fizzy drinks, red meat and cabbage were present in between 60–70% and fish, crisps and pumpkin in 50–60% of homes. Sweets and chocolates, biscuits, coffee creamer, green leafy vegetables, frozen vegetables and brown bread were present in 40–50% of homes, whilst pies and fat cakes, tinned vegetables and tinned meat were available in approximately one third or less homes (Table 3). 

#### 3.4.5. Lunchbox Practices (Learner Questions)

The majority (84%, *n* = 128) of the total sample of learners (*n* = 152) indicated that they take a lunchbox to school. Of those who did take a lunchbox to school, 67% (*n* = 86) indicated that their mother prepared it, with 20% (*n* = 25) indicating that they prepared it themselves and 13% (*n* = 17) that another person prepared it. The food items typically included in the learner’s lunchboxes are presented in Figure 8. White bread sandwiches and fruit were the most common lunch items. A fifth of the learners included cold drinks and/or fruit juice and 10% included fizzy drinks in their lunch box (in total 48% of leaners had at least one of these drinks in their lunchbox). Almost 40% of learners reported inclusion of an energy dense snack, such as crisps or sweets/chocolate. Less than 10% of learners indicated that they included cake/cookies/biscuits, yoghurt or cooked meals in their lunchboxes. 

#### 3.4.6. Household Respondent and Best Friend Likes and Dislikes of Food Items/Dishes and Snacks

Results for food items and dishes/snacks most and least liked are presented in Figure 9. The four foods most liked by household respondents were pasta dishes (25%), stews (21.7%), chicken (13%) and starches (10.9%). The least liked were stews (17.1%), pasta (12.2%), fish (11%) and vegetables, red meat and energy dense meals/snacks (all 8.5%). For learners’ best friends (as reported by learners), these were energy dense meals/snacks (39.2%), pasta (14.3%), vegetables (5.3%) and starches (5.2%), and least liked were vegetables (17.2%), stews (17.1%), starches and red meat (both 8.5%) and fish (7.3%).

For both household respondents and learners’ best friend, a SSB (with or without an unhealthy snack) was the favourite drink/snack, and salty snacks the second most favourite (Figure 10). Almost 10% of the household respondents mentioned a healthy snack as their favourite, whereas none were mentioned by learners for best friend.

## 4. Discussion

The present study makes a contribution to the emergent food environment research in LMICs. We have comprehensively reported on the home, community and school food environments of South African primary school aged learners. 

The home food environment is an important setting, influencing the development of eating behaviours and food preferences in young children. It can be conceptualised as three overlapping domains (political and economic environments, built and natural environments and socio-cultural environments) [50]. Our household survey showed that almost half of the household respondents, who were most likely to be a parent of the child, did not complete their schooling, while a third completed grade 12 and a fifth, a post-grade 12 qualification. In addition, the school principal explained that some learners lived in lower-income areas and were transported from these areas to school. Maternal education has been shown to be a predictor of children’s fruit intake [51] and low socio-economic position is consistently associated with less frequent or low intakes of fruits and vegetables [52] and higher consumption of energy dense foods [53].

Our food inventory, completed by household respondents, revealed that a variety of both healthy and unhealthy food items were available in the homes of learners at the time of the study. Items that were present in more than 70% of homes included oil, samp, pasta, rice, oats, legumes, maize meal, white bread, chicken, eggs, peanut butter, vegetables such as onions, tomatoes and potatoes and fruits. Processed meats and fizzy drinks were present in two thirds of the homes and sweets, chocolates, jam, biscuits and crisps were present in 44–54% of homes. Our Photovoice findings on foods in the home and meal components support the inventory results. When describing their meal composition during the Photovoice interviews, the meal combination that was most commonly mentioned by the learners was a combined starch, protein and vegetable dish. Less healthy meals consisting of white bread and slap chips only, and white bread and polony were also described during the interviews. Findings from systematic reviews, largely from cross-sectional studies, have concluded that children’s fruit and vegetable (F&V) intakes are positively related to home availability of these foods [54,55] and that the accessibility of healthful foods in the home has been inversely associated with children’s total energy and fat intake [56]. Having fewer unhealthy items in the home may also encourage children and adolescents to consume more F&V and lean meats [57]. This is important given that a number of national surveys in South Africa have shown that F&V intake is below recommendations in all age groups [58,59]. For example, results from the South African Provincial Dietary Intake Survey in 1 to >10-year-old children showed that salty snacks contributed a third of total energy intake (maize porridge contributed most to total energy intake), while also being the top source of total fat and saturated fat. Granulated sugar (e.g., sugar added to hot drinks) was found to be the third highest contributor to total carbohydrate intake [49].

The majority of household respondents in our study indicated that they encouraged their children to eat F&V and brown bread, liked the taste of F&V and were willing to try other types of foods. They also indicated that they eat the foods that they want their children to eat. A number of studies have reported positive cross-sectional relationships between maternal or parental F&V intake and children’s consumption [55,60,61]. Further, a study comparing the influence of parental role modelling with parental dietary intake on children’s diet quality, demonstrated that parental modelling of healthy eating was particularly important for influencing children’s diet quality [62]. Other studies have demonstrated that household food rules are positively associated with children’s dietary quality [48] and children’s dietary fat intake [63], but not associated with children’s SSB consumption [64]. Interestingly, most household respondents in our study reported the existence of household rules around the consumption of crisps, fat cakes/doughnuts, sweets and chocolates. Speculatively, it is possible that some of the drinks/snacks liked most (e.g., SSBs and salty snacks) and food items disliked (vegetables) by our household respondents may have influenced their intention to encourage and model healthy eating behaviours and to set household food rules. 

Approximately half of the household respondents thought that what their child’s friends eat influences what their child eats/drinks. At the top of the list of foods learners thought their best friend liked most were energy dense meals or snacks, while vegetables were at the top of the list of the most disliked foods. Learners thought the snacks that their best friend liked the most were SSBs (with or without an unhealthy snack) and crisps/salty biscuits, which is interesting given that peer influence increases during childhood and adolescence, and that friendship groups may play a role in determining eating patterns [65]. Two studies have shown the presence of peers and friends at eating occasions increases adolescent energy intake and the likelihood of meal and snack consumption [66,67]. In contrast, support for healthy eating in the form of friends eating healthy foods together, discouragement of the consumption of junk food and encouragement of the consumption of healthy foods by best friends [67] has been associated with a change in vegetable consumption [68]. It is possible, therefore, that our learners’ friends may have influenced an unhealthy meal and snack option choice, although much of the literature has shown that modeling of healthy eating by best friends was not associated with better eating behaviours [69,70]. 

A variety of aspects of mealtime structure, including eating together as a family, watching television during meals and where the meal is prepared, appear to influence children’s dietary behaviours [71]. Our Photovoice photographs and information provided in the interviews revealed that eating chips and sweets in front of the TV, at the table or in the bedroom with family members was present. Our household survey results also showed that eating a meal and/or snacks together whilst watching TV was common behaviour. Studies from the US and Australia have shown that high TV use during mealtime was associated with children’s increased energy intake, specifically in the form of snacks and high-energy drinks [72,73]. Eating a home-prepared meal together as a family did not emerge as themes in the Photovoice results, although the majority of household respondents reported that family members ate supper together and consumed a home-cooked meal regularly. Importantly, frequent family meals have been shown to be associated with improved F&V intake in adolescents [73].

Food availability is a key dimension of the community food environment [74], and the household survey showed that Spaza shops, General Dealers and supermarkets were the food outlets most commonly frequented by household members. Our Photovoice findings revealed that staples such as bread, maize and spices, purchased at supermarkets, were consistent with the household survey where a quarter of respondents indicated that food purchases occurred at supermarkets on most days. Whilst supermarkets are considered to stock better quality and more variety of foods [75], possibly at a lower cost, the reliance on public transport in getting to these shops, carrying heavy groceries home or depending on others for private vehicle use, are considered barriers to accessing these larger food outlets [76]. Limited access and availability (commonly described as ‘food deserts’ where healthy foods such as F&V are insufficient [74]), can therefore mean less shopping trips. Thus, location and accessibility to stores is an important determinant in food purchasing behaviour [75,76]. 

Interestingly, although the learners did not mention or photograph street vendors as a place to buy snacks or beverages, the principal expressed concern about vendors located close to the school and that learners often bought unhealthy snacks from these vendors on their way to school. Street vendors in low income areas, close to schools, is not uncommon [19]. A number of studies in poorly resourced schools in South Africa have reported that unhealthy snacks and beverages are the main items sold by street vendors. For example, De Villiers and colleagues showed in a survey of 100 schools in urban and rural areas in the Western Cape, that the most common items sold by street vendors were sweets, crisps, ice lollies, doughnuts, hot dogs/burgers and fat cakes (fried dough balls) [18]. 

Informal home shop vendors were commonly cited by learners in the Photovoice interviews as places of purchase by either themselves and/or household members. Items bought from these vendors included mostly snacks and SSBs; however, grocery items (e.g., F&V and canned foods) were also mentioned. Although learners did not mention informal home shop vendors as having less variety and fewer healthy choices available than other food outlets, a number of their photographs depicted the vendors stocked with crisps, chocolates and fizzy drinks rather than fresh F&V and a selection of grocery items. Findings in countries such as the US have also shown that local stores were predominantly stocked with snacks and SSBs [77,78], while the availability of fresh produce is described as unreliable and sporadic [76,79]. This often leads to an increased consumption of these convenience foods [79]. The in-store availability of healthy and unhealthy options has often been identified as playing a key role in food purchasing decisions, particularly amongst lower-income populations, and the reliance on unhealthy snacks and SSBs contributes towards the increased vulnerability to the nutrition transition in Africa [80]. 

It emerged from the Photovoice results that home shop vendors were conveniently located in physical proximity to the homes of learners, often mentioned as close by or next-door. Walkability has been identified as a key priority for low-income population groups in the UK and the US, without access to cars [81,82]. Informal retail stores, e.g., General Dealers and Spaza stores, were also places where household respondents chose to shop on most days, most likely due to the close location to their home. Notably, the school principal raised one influencing factor, broader than the food environment, namely neighbourhood safety concerns. He described the learner’s neighbourhoods as unsafe, with gun related violence incidents, influencing the learner’s or their parent’s ability to access shops, which in turn limits food choice. Other studies from the US have also identified personal safety as a determinant of shopping location [83], with people choosing to avoid stores where they had heard of violence occurring [83].

A tuckshop operated daily at our study school, where learners bought snacks and drinks. An inventory of the tuckshop revealed that only unhealthy snack foods were available for sale and, according to our purchase observation, the most popular snack items bought were sweets, chocolates and biscuits. Similarly, two other South African studies reported learners purchasing sweets, chocolates and chips from the school tuckshop [18,20], whilst Wiles et al. found the most popular items sold were pies and iced lollies [84]. Globally, children consume almost 40% of their daily recommended energy intake from EDNP foods [85,86] and the availability of these types of foods at school tuckshops undermines compliance with food-based dietary guidelines [87]. Tuckshops have been identified by the World Health Organisation as an effective setting to improve children’s nutrient intakes [88]. Recommendations are to limit unhealthy snacks sold at school tuckshops and for healthier items to be made available. In South Africa, only 8% of schools with a tuckshop have been reported to have a policy for operating purposes [18]. Household respondents in our study were generally in agreement that the types of snacks/foods that children see for sale at tuckshops influence what they eat, highlighting the importance of tuckshop policies in limiting unhealthy snacking. Furthermore, learners were aged 10 to 13 years old, which is a time when children begin to develop greater autonomy in decision making [89] under scoring the importance of healthy options early in life.

The principal spoke of a desire to limit the number of unhealthy snacks and drinks, and his success in reducing the availability of some of these foods, although they were not replaced with healthier options. He also emphasised the importance of the tuckshop profits, which financed the purchase of stationary and maintaining the school premises. Other studies have also revealed the barriers that schools must overcome to sell healthier foods, e.g., a fear of losing income through selling these foods [18], children’s preference for unhealthy snacks [87,88] and the higher cost of healthier foods [20,90].

South African schools encourage their learners to carry lunchboxes, with some implementing a policy on limiting ‘junk food’ and including fruit and drinks [19,91]. According to our learners, most took a lunchbox to school and their mother prepared the lunchbox contents. Common lunchbox food items were white bread sandwiches (with polony), fruit, crisps and sweets/chocolates. In contrast, another South African study found only a quarter of learners took a lunch box to school, although the packed items were similar: bread/sandwiches and meat egg, fish, polony and porridge [19]. The majority of our household respondents agreed that taking a lunchbox to school influences what a child eats. Findings from Abrahams et al. reported that younger South African students were more likely to bring a lunchbox to school than older students and those students who did not bring a lunchbox to school were more likely to purchase less healthy snacks from the tuckshop [92].

Regarding the school garden, the principal explained that the grown vegetables were used to supplement the meals prepared for the 50% of learners who participated in the NSNP. Prior to the establishment of the garden, meals consisted of soya and rice, and a fortified, sweetened milk drink. The lack of fresh F&V concerned the principal, and he believed that the addition of fresh garden vegetables to the meals was beneficial for the learners and financially advantageous for the school. A large proportion of South African schools do not comply with the mandated serving/day of F&V [93] and studies have shown that children’s preference, consumption and knowledge of F&V can be positively affected through school gardens [94,95]. Previous evidence has revealed that lack of space and unsuitable grounds were key barriers for South African school vegetable gardens [18]. However, our study school successfully cultivated a hydroponic garden in a small area of the school yard, underscoring their commitment to providing fresh vegetables to their pupils.

The NSNP implemented in our study school not only focused on feeding the learners, but also included an educational component on healthy eating for all learners. Schools are an ideal setting for child-focused initiatives, as they offer continuous and intensive contact with children for prolonged periods [96]. Two studies involving curriculum-based nutrition education approaches have demonstrated significant improvements in children’s F&V consumption [97] and a reduction in total energy intake [98]. The principal also emphasised the importance of the teacher’s involvement in role modelling positive eating behaviour. Our household respondents also considered this as important, with one third noting that what teachers eat/drink influences what their child chooses to eat/drink. The concept that teachers are not only providers of education is supported by two studies which revealed that teachers, through positive role modelling, can influence learners at school to eat well [99,100] and to be highly motivated [12].

### 4.1. Strengths and Limitations

The present study has a number of strengths which include the use of qualitative and quantitative research methods, which provided comprehensive and contextual insight into the home, community and school environments of learners in an urban setting in South Africa. Data were drawn from a sample of demographically diverse learners and their households. Several under-examined dimensions of the South African home food environment were investigated. It is important to also acknowledge the limitations of the study. Data were drawn from one school and thus the generalisability of this study is limited. The self-reported data are subject to socially desirable response bias or misreporting, and learner purchasing behaviour may have been influenced during the tuckshop observation.

### 4.2. Implications for Future Directions

Firstly, the foods commonly available in learners’ homes comprised a combination of food groups, namely: refined carbohydrates; fats and oils; chicken and processed meats; vegetables, legumes; sugary foods. Collectively, these foods are broadly termed the ‘Western diet,’ characterised by high amounts of energy, fat, sugar and sodium and are associated with obesity and diet-related chronic disease [101]. Secondly, the informal food outlets, primarily shopped at by the learners and/or household respondents, may be a key influencing factor in the purchasing of these types of foods. As noted, many of these food outlets exist in low-income informal settlements, in close proximity to schools and/or homes, limiting other healthy food options. Thirdly, the school tuckshop provided a number of unhealthy, processed food items, which promotes the notion that these types of foods can be consumed on a daily basis rather than occasionally [57]

Based on our findings, several recommendations can be made to better understand the mechanisms underlying the relationship between food availability and accessibility, consumption and nutritional status in the South African context. An exploration of dietary patterns would help elucidate important diet and disease relations and provide dietary advice that is understood by the community. To support households, understanding the facilitators (e.g., convenience) and barriers (e.g., perceived cost, low socio demographic factors) to making a greater variety of healthy foods available in homes is required. A similar, wide-ranging school-based approach could also be implemented. Comparing price with accessibility would provide valuable insights, as food prices, measured by either costing shopping baskets of commonly purchased foods or by ranking food outlets by price of products sold, remain relatively understudied in South Africa. Less studied dimensions such as the preference and desirability towards convenience vs. healthy foods, and the role these personal factors play in food-related behaviours, should also be explored. Practical strategies to assist informal food outlets to provide healthier food options while maintaining profitability could also be an area of research attention. Finally, although South African school tuckshops have been the focus of a number of cross-sectional and intervention studies, limiting the provision of unhealthy school tuckshop food and snacks still appears to be a challenge. Further support and guidance for schools to establish novel strategies to implement healthy tuckshop policies, whilst still generating an income to cover school operational costs, is required. We also recommend that sustainable vegetable gardening and the addition of the grown vegetables to the school meals be considered when designing interventions aimed at improving the school food environment and nutritional status of learners.

## 5. Conclusions

Overall, it appears that availability and accessibility to cheap, convenient and desirable foods, coupled with the economic constraints, limits opportunities for healthier alternatives. Improving our understanding of the South African food environment will be critical to the design of effective and targeted interventions and policies aimed at improving public heath nutrition and the burden of malnutrition. Findings from our study have implications for informing further research into families living in historically disadvantaged and informal settlements in urban areas, which will aid the development of strategies to support healthy food environments for South African children.

## Figures and Tables

**Figure 1 nutrients-13-02043-f001:**
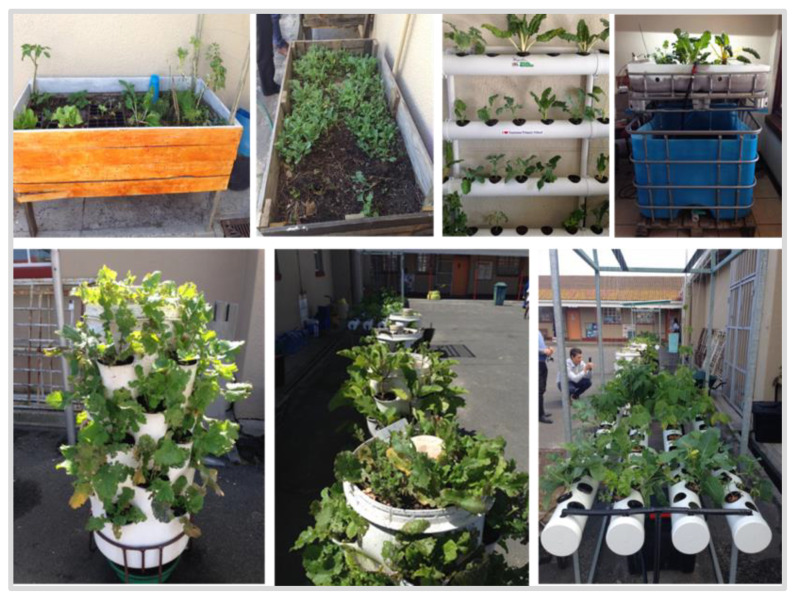
Photographs of vegetable gardening at the participating school.

**Figure 2 nutrients-13-02043-f002:**
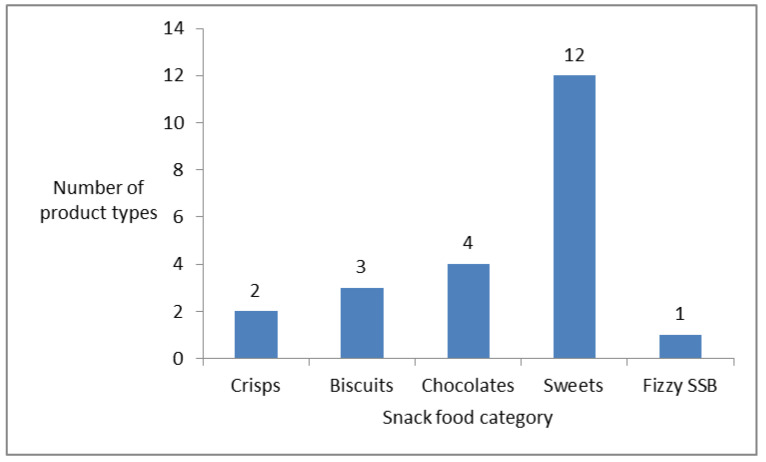
Total number of product types available for each snack food category at the school tuckshop in the study school.

**Figure 3 nutrients-13-02043-f003:**
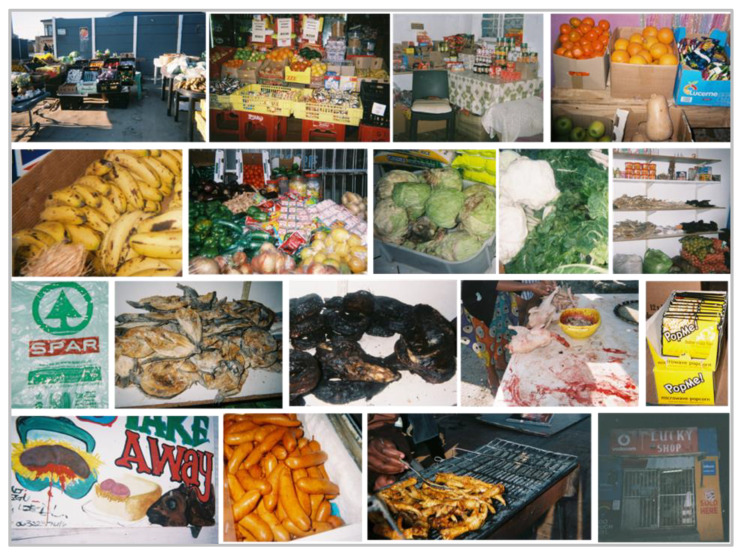
Photographs taken by grade seven learners of food shops, informal vendors and take-away options in their community environment.

**Figure 4 nutrients-13-02043-f004:**
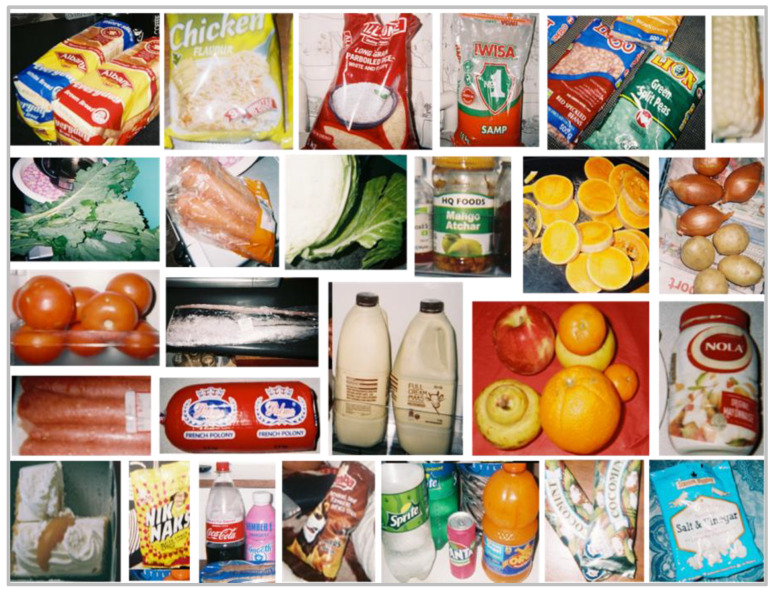
Photographs taken by grade seven learners of foods and snacks available in their homes.

**Figure 5 nutrients-13-02043-f005:**
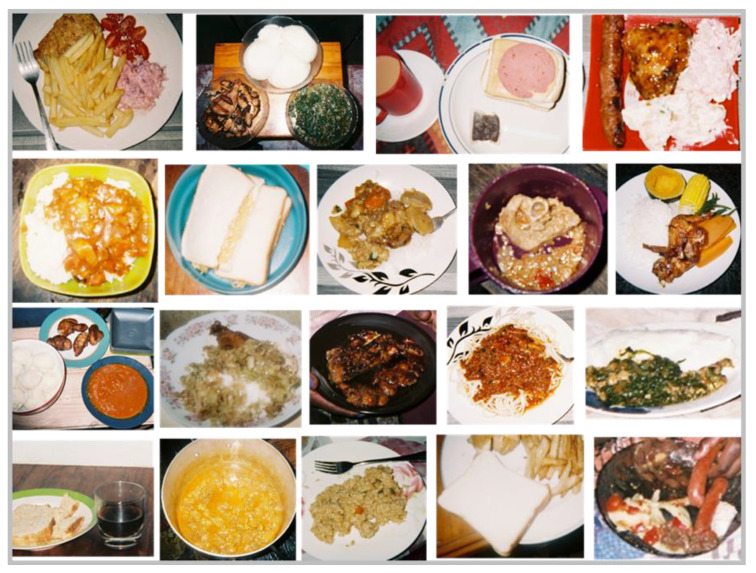
Photographs taken by grade seven learners of meals served in their homes.

**Figure 6 nutrients-13-02043-f006:**
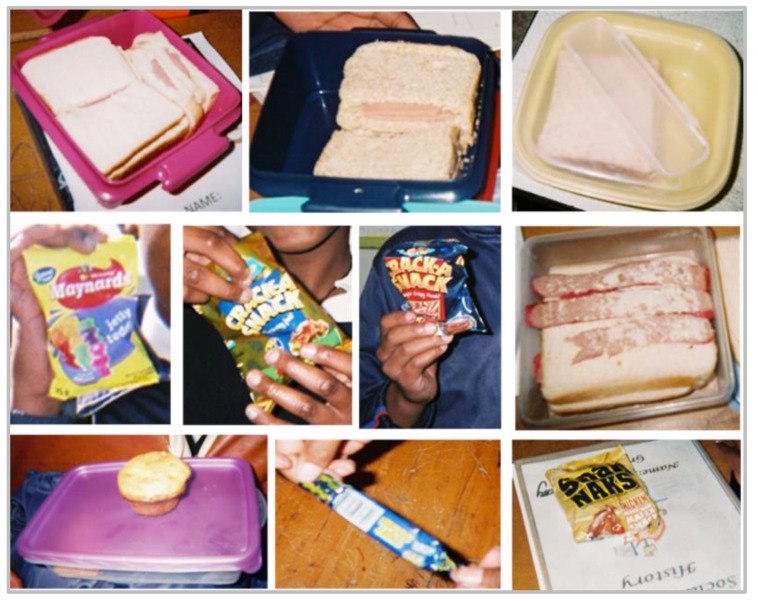
Photographs taken by grade 7 learners of their lunch boxes and snacks eaten at school.

**Figure 7 nutrients-13-02043-f007:**
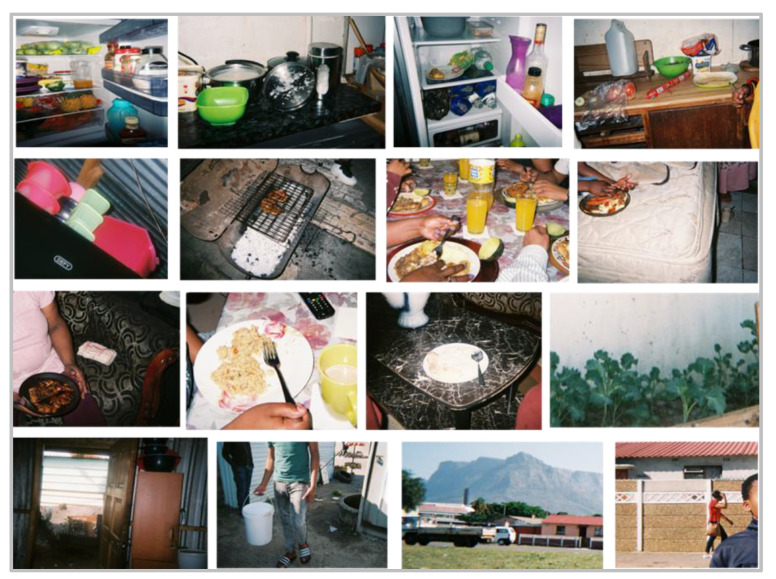
Photographs taken by grade seven learners of household cooking and food storage facilities, meal eating occasions/locations and areas outside their homes, including a vegtable garden where “covo” was grown.

**Figure 8 nutrients-13-02043-f008:**
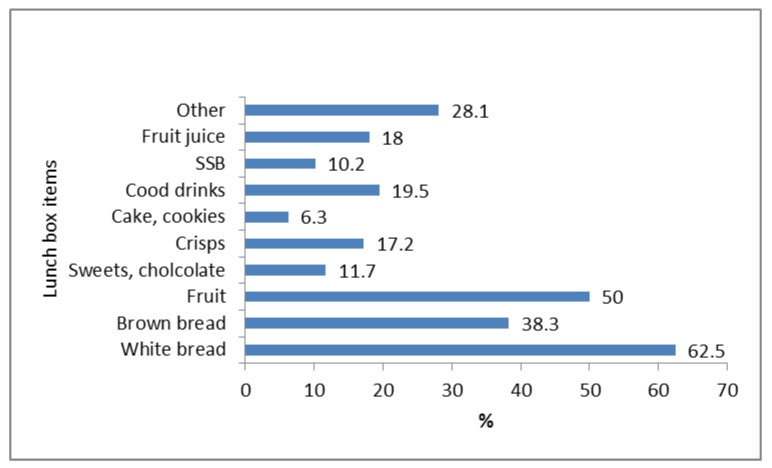
Food/snack/drink items learners indicated they have in their lunchboxes most of the time (*n* = 128).

**Figure 9 nutrients-13-02043-f009:**
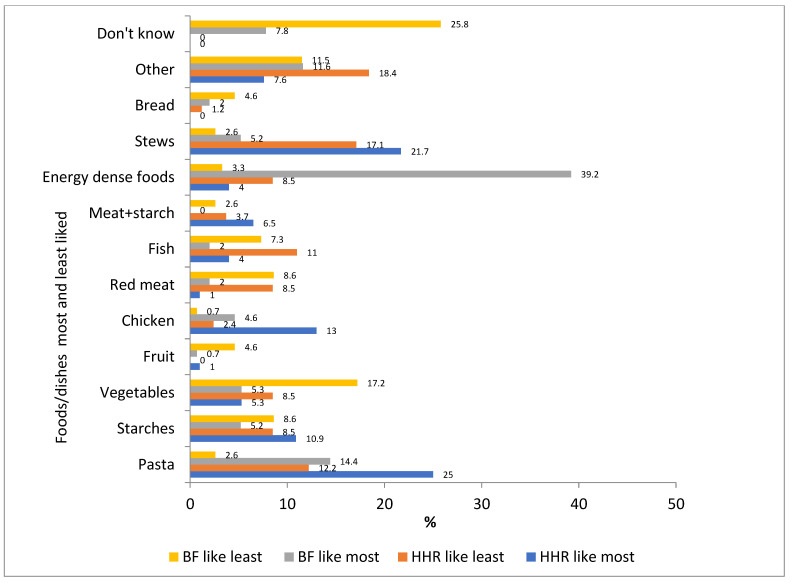
Foods/dishes most and least liked by household respondents (HHR *n* = 92) and learners’ perceptions of their best friend’s most liked and least liked foods/dishes (BF *n* = 153).

**Figure 10 nutrients-13-02043-f010:**
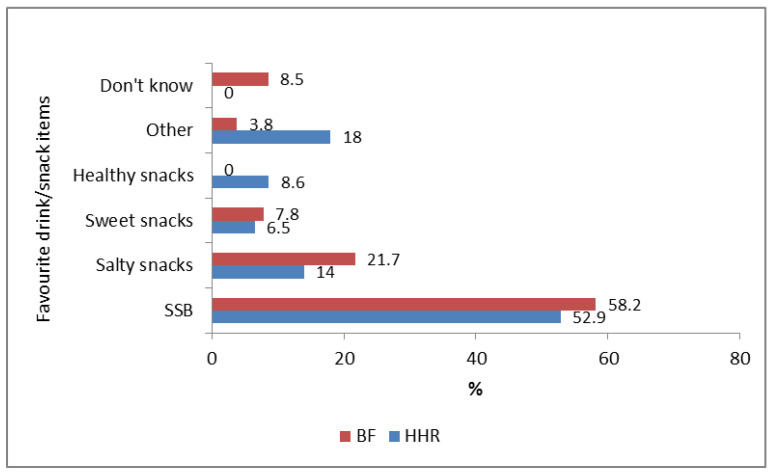
Most favourite snack/drink of household respondents (HHR *n* = 92) and learners’ perception of their best friend’s favourite snack/drink (BF *n* = 153).

**Table 1 nutrients-13-02043-t001:** Socio-demographic profile of household respondents and households.

Household Respondent		Household	
Relationship to Learner *n* (column%)	*N* = 101	Food Security *n* (Column%)	*N* = 100
Father	25(24.5)	No Hunger	55(55)
Mother	69(68.3)	At risk of hunger	24(24)
Aunt	2(2.1)	Hunger	21(14)
Grandmother	4(4)	Number of adults	*n* = 95
Sister	1(1.1)	Median (IQR	2(2:4)
Age	*n* = 94	Number of children	*n* = 101
Mean (SD) years	39.5 (9.3)	Median (IQR)	2(2:3)
Marital Status *n* (column %)	*n* = 101		
Married	63(62.3)		
Divorced	4(4)		
Single	26(25.7)		
Widowed	59(5)		
Other	3(3)		
Education level *n* (column %)	*n* = 100		
Primary or less	9(9)		
Some High School	38(38)		
Grade 12 only	30(30)		
Grade 12 +	23(23)		

Note: *n* varies due to missing values.

**Table 2 nutrients-13-02043-t002:** Household eating and food purchasing behaviour as reported by the household respondent.

Household Eating Behaviour	≤2/wk	3–4/wk	≥5/wk	Household Influences on a Child’s Eating Behaviour	Yes *	Have Household Rules Relating to:	Yes *	Factors that Respondent Thinks Influences What Child Eats	Yes *	Where the Family Purchases Food	≤2/wk	3–4/wk	≥5/wk
*n* = 102	%	%	%	*n* = 102	%	*n* = 101	%	*n* = 100	%	*n* = 98	%	%	%
Eat supper together as a family	21.5	12.7	65.7	Respondent eats food he/she wants child to eat	70.3	Fizzy drinks	78.4	Child’s knowledge of healthy eating	78.2	Spaza shop	66.3	12.2	21.5
Eat breakfast together as a family	61.3	12.9	25.7	Respondent encourages child to eat vegetables	89.2	Fat cakes, doughnuts, slap chips	75.4	Whether child takes a lunch box to school	73.3	Cafe	81.9	8.5	9.6
Family eats a meal in front of the TV	27	14	59	Respondent encourages child to eat fruit	94.1	Sweets, chocolates	77.2	What is sold at school tuck shop and other food outlets	63	General dealer	60.2	11.2	28.5
Family eats snacks in front of the TV	41.4	16.1	42.4	Respondent encourages child to eat brown/whole wheat bread	70.6	Sweet biscuits, tarts, cakes	74.3	Child’s body image	66	Supermarket	56	18	25.2
Family eats a home cooked meal	13.8	13.8	72.2	Respondent encourages child to eat all food on his/her plate	81.4	Take-out foods	66.3	What people living with a child eat/drink	54.1	Whole sale	84	4.1	9.2
Family eats vegetables with a meal	45	29	26	Respondent encourages child to eat at a table	78.2	Crisps	66.7	Advertisements, TV and billboards	48.5				
				Cultural factors determine what the family eats	50	Pies, samosas, sausage rolls	65.7	What a child’s friends eat/drink	46.5				
						Sugar	64.7	Child’s parents’ working hours	41.4				
								What a child’s school educators eat/drink	31.3				

* The balance of household respondents reported ‘No’.

**Table 3 nutrients-13-02043-t003:** Foods and beverages available in the household as reported by the household respondent (*n* = 97).

Foods and Beverages	Y (%)	Foods and Beverages	Y (%)
Samp (dried corn kernels), pasta, roti	96.8	Jam	67.7
Oil	93.6	Cheese	67.3
Sugar	91.5	Fizzy drinks	66.6
Fats	90.5	Red meat	60.2
Onions	89.4	Cabbage	60
Dairy	88.4	Fish	59.1
Chicken	87	Crisps	53.7
Potatoes	85.2	Pumpkin	50.5
Eggs	81.7	Biscuits	49.4
Tomatoes	81	Green leafy vegetables	47.3
Peanut Butter	80.6	Creamer	44
Oats cereal	78.8	Sweets and chocolates	44
Fruit	77.8	Frozen vegetables	42.1
White bread	75	Brown bread	41
Legumes	74.1	Pies, fat cakes	35.4
Maize meal	73.6	Tinned vegetables	30.5
Carrots	68.4	Organ meat	23.6
Viennas	67.7	Tinned meat	21.7

Definitions: Viennas—processed meat sausage made from pork or chicken; creamer-ultra-processed powdered coffee creamer; samp-dried corn kernels boiled until tender.

## Data Availability

The data presented in this study are available on request from the corresponding author pending ethical approval from the Faculty of Health Sciences Human research Ethics Committee, University of Cape Town. The data are not publicly available due to privacy restrictions.

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
