# Peer review of "The Food Environment of Primary School Learners in a Low-to-Middle-Income Area in Cape Town, South Africa"

_nutrients, 2021, doi:10.3390/nu13062043_

Round 1

Reviewer 1 Report

A brief summary (one short paragraph) outlining the aim of the paper and its main contributions.

  • This paper aims to expand the food environment literature to include low-and-middle income countries, specifically in South Africa children living in historically disadvantaged and informal settlements in urban areas. The authors state that while some work has been done in South Africa, it has occurred in either home, community, or school food environments and has yet to capture all three within a single population. This paper contributes to the field by taking a case study approach to present on all three levels – home, community, and school – among primary school aged learners.

Broad comments highlighting areas of strength and weakness. These comments should be specific enough for authors to be able to respond.

Strengths

  • Methods appear to be comprehensive and controlled.
  • Addresses a gap in food environment research

Weaknesses:

  • The introduction should include more background on why food environments should be investigated in children specifically. What is the current health status of South African children? Is there high childhood obesity? Are there common nutrient deficiencies? Are there predictions that unhealthy trends observe in childhood/adolescence will carry over into adulthood, and if yes, what are the implications of this? I don’t see any argument as to the value and importance of this work in children/adolescents.
  • The terms “children” and “adolescents” are used interchangeably throughout, though they represent distinct age groups and stages of development. It would be valuable to indicate that the grades included in this analysis represent a transition between the two stages which is often characterized by increased independence and increased autonomy when it comes to food purchasing and consumption. It is an important time for establishing healthy eating behaviors, as they are often carried into adulthood.

Specific comments referring to line numbers, tables or figures. Reviewers need not comment on formatting issues that do not obscure the meaning of the paper, as these will be addressed by editors.

  • Line 23: I don’t understand the use of colons here. Should these be semi-colons?
  • Line 44: Remove comma after “intake”
  • Lines 74-79: It seems like peer influences would also be present within the school environment, but it is only included here within the community environment. Was this decision based on previous literature that there is no peer influence within schools? This comes up in Lines 218-219 with the questions about which foods their best friend liked least/most, which sounds like peer influence.
  • Line 84: Is there a stigma associated with NSNP participation? Is it clear which learners receive NSNP and which do not? Could this influence learner’s willingness to participate? This info may not belong here, but could be interesting to note when discussing the role of NSNP in the school food environment.
  • Line 98: What grades were taught at this primary school? This differs by country and even region within countries, so would be good to clarify the grades taught as well as age demographics (you mention equal distribution of boys and girls, but what ages are typically represented at a primary school in South Africa?)
  • Line 107: Remove comma after “transported”
  • Line 114: Change to “dietitian.”
  • Line 118: Remove comma after “panel”
  • Line 120: It is mentioned that the interview guide included questions on the learning profile and general health of learners. I am wondering if this profile was for all students at the school or just those in grades 4-7, since those were the grades involved in the tuckshop observation, or grades 5-7, since those were the grades involved in the household interviews? I imagine there are many differences in learner profile and general health of learners, for example between learners in grades 1-3 and those in grades 4-7. Would be good to know if the principal was providing information for the school as a whole or just for the grades captured in the tuckshop observation and/or household interviews.
  • Line 158: Remove comma after “2019”
  • Line 177: Remove comma after “NP”
  • Line 309: Remove comma after “items”
  • Lines 319-325: I’m not sure about how this section directly relates to food environment. Being that it is information attained from one interview, and seems to be based on speculation, it may not be needed, especially since alcohol and drug use can be a sensitive and stigmatizing topic.
  • Line 392: Should “Figure 7” be changed to “Figure 4: row 1, column 2”?
  • Lines 445-453: Why are direct quotes not italicized here as they are in previous paragraphs?
  • Line 456: Change to “…who were most likely to be parents of the learners”
  • Line 461: Remove comma after “households”
  • Table 1: Formatting seems to have gotten messed up, I’m not sure how the columns are labeled. This table is difficult to read. I suggest left alignment and then indenting rows under each section.
  • Line 478: Add comma after “fruit”
  • Line 485: Change to past tense
  • Line 486: I don’t understand this sentence, it seems like it is a combination of two things and repeats what is in line 487?
  • Table 2: Formatting again seems to be messed up, making this table difficult to read. Suggest left justify.
  • Line 524: Is this meant to say “Less than 10% of learners indicated that they included cake/cookies/biscuits…”?
  • Line 545-552: Will this analysis be performed in subsequent papers? If you’re going to make speculation about the role of maternal education and low SES in children’s dietary intake, why not see if these factors also influence children’s food environments? It seems like you have the data for this. I understand it may not belong in this paper, but it does seem like a future direction. Perhaps mention in the “Implications for future directions” section?
  • Line 569: Define F&V
  • Line 592: Change to past tense
  • Line 603: A positive or negative change in vegetable consumption? Give directionality.
  • Line 604: Change to “It is possible, therefore, that our learners’ friends…”
  • Line 611: Remove comma after “interviews”
  • Line 614: Clarify that this is high TV use during meal times, not just in general.
  • Line 632: Why is there this discrepancy between principal’s emphasis on influence of street vendors and low frequency of actual learner purchases from street vendors? Curious about possible explanations.
  • Lines 648-653: If you are going to include comparisons to US data, you may consider citing more recent work by Gittelsohn and colleagues on neighborhood corner store food environments in inner city Baltimore and food purchasing habits of adolescents.
  • Line 735: Change to “Data were…”
  • Line 736: Response bias is mentioned, but what about potential change in purchasing behaviors at the tuckshop due to observation?
  • Line 763: Remove comma after “tuckshops”
  • Line 767: Add comma after “costs”
  • Line 769: Remove comma after “meals”
  • Line 773: Remove comma after “constraints”
  • Lines 774-776: Make specific to children’s food environment, or indicate that analysis of children’s food environments is an important first step in characterizing food environments as a whole.

Author Response

The food environment of primary school learners in a low-to-middle-income area in Cape Town, South Africa

We would like to thank the reviewers for their time and attention spent reviewing and for their helpful commentary.

Reviewer 1

Comment 1

The introduction should include more background on why food environments should be investigated in children specifically. What is the current health status of South African children? Is there high childhood obesity? Are there common nutrient deficiencies? Are there predictions that unhealthy trends observe in childhood/adolescence will carry over into adulthood, and if yes, what are the implications of this? I don’t see any argument as to the value and importance of this work in children/adolescents.

Response 1

Thank you for your comment.

Page 1, Lines 37-40 now reads:

In South Africa adolescents especially urban females, were burdened by overweight and obesity with prevalence increases of 6% in boys and 7% in girls between 2002-2008 [3].

Comment 2

The terms “children” and “adolescents” are used interchangeably throughout, though they represent distinct age groups and stages of development. It would be valuable to indicate that the grades included in this analysis represent a transition between the two stages which is often characterized by increased independence and increased autonomy when it comes to food purchasing and consumption. It is an important time for establishing healthy eating behaviors, as they are often carried into adulthood.

Response 2

Thank you for your comment. We have only used the term“adolescents’ on page 18 line 568 and page 19 line 690 in reference to findings highlighted from other papers. However, we have added on page 22 lines 1018-1020:

Furthermore, learners were aged 10-to-13-years-old, which is a time when children begin to develop greater autonomy in decision making[91] underscoring the importance of healthy options early in life.

Comment 3

Line 23: I don’t understand the use of colons here. Should these be semi-colons?

Line 44: Remove comma after “intake”

Line 107: Remove comma after “transported”

Line 114: Change to “dietitian.”

Line 118: Remove comma after “panel”

Line 158: Remove comma after “2019”

Line 177: Remove comma after “NP”

Line 309: Remove comma after “items”

Line 392: Should “Figure 7” be changed to “Figure 4: row 1, column 2”?

Lines 445-453: Why are direct quotes not italicized here as they are in previous paragraphs?

Line 456: Change to “…who were most likely to be parents of the learners”

Line 461: Remove comma after “households”

Line 478: Add comma after “fruit”

Line 485: Change to past tense

Line 569: Define F&V

Line 592: Change to past tense

Line 603: A positive or negative change in vegetable consumption? Give directionality.

Line 604: Change to “It is possible, therefore, that our learners’ friends…”

Line 611: Remove comma after “interviews”

Line 735: Change to “Data were…”

Line 763: Remove comma after “tuckshops”

Line 767: Add comma after “costs”

Line 769: Remove comma after “meals”

Line 773: Remove comma after “constraints”

Response 3

Please refer to the manuscript for edits.

Comment 4

Lines 74-79: It seems like peer influences would also be present within the school environment, but it is only included here within the community environment. Was this decision based on previous literature that there is no peer influence within schools? This comes up in Lines 218-219 with the questions about which foods their best friend liked least/most, which sounds like peer influence.

Response 4

Thank you for your comment. We agree that peer influences on learners’ eating behaviours could also be observed in the school environment. Our questions about peers likes and dislikes influenced were part of the leaner questionnaire and thus were grouped in the community environment.

Comment 5

Line 84: Is there a stigma associated with NSNP participation? Is it clear which learners receive NSNP and which do not? Could this influence learner’s willingness to participate? This info may not belong here, but could be interesting to note when discussing the role of NSNP in the school food environment.

Response 5

We are not aware of any South African studies which have explored learners experience of stigma when participating in the NSNP. A study by Faber et al. found that parents were positive towards the provision of the school meal but the authors did not explore learner’s attitudes or experiences. The decision to receive meals as part of NSNP is made by the parents. We agree that learners maybe made to feel embarrassed about receiving meals which in turn, could influence their willingness to participate. We thank you for your suggestion and think this is an interesting topic which may be more suited to another paper in the future.

Comment 6

Line 98: What grades were taught at this primary school? This differs by country and even region within countries, so would be good to clarify the grades taught as well as age demographics (you mention equal distribution of boys and girls, but what ages are typically represented at a primary school in South Africa?)

Response 6

We have added more details about the grades and age range of the learners.

Page 3 Lines 189-192:

The total number of learners across all grades (grades 1-to-7) in the school at the time of the study was 668 (typically aged 6-to-13-years), with an approximate equal distribution of boys and girls.

Comment 7

Line 120: It is mentioned that the interview guide included questions on the learning profile and general health of learners. I am wondering if this profile was for all students at the school or just those in grades 4-7, since those were the grades involved in the tuckshop observation, or grades 5-7, since those were the grades involved in the household interviews? I imagine there are many differences in learner profile and general health of learners, for example between learners in grades 1-3 and those in grades 4-7. Would be good to know if the principal was providing information for the school as a whole or just for the grades captured in the tuckshop observation and/or household interviews.

Response 7

Thank you for your comment. Please note that we refer to ‘learner’ and not ‘learning’ profile in our methods (please refer to the Principal interview guide that has now been included as a supplementary document).  The principal interview was designed to gather information primarily about the school food environment and to a lesser extent about the learners, as outlined in the methods. The learner profile included questions about the number of learners at the school, gender distribution, where the learners lived, how they got to school and circumstances at home as indicated in the interview guide and reported results. The question regarding learner health was formulated generically but the principal did not elaborate on health per se when queried. Instead, he talked about the value of the school feeding programme which was recorded in the interview results. 

Comment 8

Lines 319-325: I’m not sure about how this section directly relates to food environment. Being that it is information attained from one interview, and seems to be based on speculation, it may not be needed, especially since alcohol and drug use can be a sensitive and stigmatizing topic.

Response 8

Within the context of the key informant interview methodology, mentioning this issue, although it is the opinion of one person, is appropriate. Insights from the principal about learner’s family structure are important, particularly those from low socio-economic areas of South Africa, as it provides context around the home food environment. We appreciate your comment but prefer to keep these lines unchanged.

Comment 9

Table 1: Formatting seems to have gotten messed up, I’m not sure how the columns are labeled. This table is difficult to read. I suggest left alignment and then indenting rows under each section.

Response 9

Thank you. The editing team will amend accordingly.

Comment 10

Line 486: I don’t understand this sentence, it seems like it is a combination of two things and repeats what is in line 487?

Response 10

Please refer to the edited sentence page 14 lines 695-699.

Half or fewer of the respondent agreed that what other people living with a child eat and drink; what children see in advertisements, on TV and on billboards; what children’s friends eat and drink; children’s parents’ working hours; and what children’s teachers eat and drink influence what they eat (Table 2)”.

Comment 11

Table 2: Formatting again seems to be messed up, making this table difficult to read. Suggest left justify.

Response 11

Thank you. The editing team will amend accordingly.

Comment 12

Line 524: Is this meant to say “Less than 10% of learners indicated that they included cake/cookies/biscuits…”?

Response 12

Please refer to the edited sentence page 16 lines 735-737.

Less than 10% of learners indicated that they included cake/cookies/biscuits, yoghurt or cooked meals in their lunchboxes.

Comment 13

Line 545-552: Will this analysis be performed in subsequent papers? If you’re going to make speculation about the role of maternal education and low SES in children’s dietary intake, why not see if these factors also influence children’s food environments? It seems like you have the data for this. I understand it may not belong in this paper, but it does seem like a future direction. Perhaps mention in the “Implications for future directions” section?

Response 13

Please refer to the edited sentence page 23 lines 1241-1242

To support households, understanding the facilitators (e.g. convenience) and barriers (e.g. perceived cost, (socio demographic factors) to making a greater variety of healthy foods available in homes is required.

Comment 14

Line 614: Clarify that this is high TV use during meal times, not just in general.

Response 14

Please refer to the edited sentence page 20 Lines 854-856

Studies from the US and Australia have shown that high TV use during mealtime was associated with children’s increased energy intake, specifically in the form of snacks and high-energy drinks.

Comment 15

Line 632: Why is there this discrepancy between principal’s emphasis on influence of street vendors and low frequency of actual learner purchases from street vendors? Curious about possible explanations.

Response 15

Thank you for your comment. It’s important to note that our reported results are not quantitative findings, thus purchases were frequently photographed do not reflect actual quantitative frequency.

Comment 16

Lines 648-653: If you are going to include comparisons to US data, you may consider citing more recent work by Gittelsohn and colleagues on neighborhood corner store food environments in inner city Baltimore and food purchasing habits of adolescents.

Response 16

Please refer to page 20 lines 854-856

We have included reference:Christiansen,KMH.; Qureshi,F.;Schaible,A.;Park,S.;Gittelsohn, J. Environmental Factors That Impact the Eating Behaviors of Low-income African American Adolescents in Baltimore City.JNutr Educ Behav. 2013;45:652-660

Findings in countries such as the US have also shown that local stores were predominantly stocked with snacks and SSBs [78,79], while the availability of fresh produce is described as unreliable and sporadic [77,80].

Comment 17

Line 736: Response bias is mentioned, but what about potential change in purchasing behaviors at the tuckshop due to observation?

Response 17

Please refer topage 23 lines 1221-1223:

The self-reported data are subject to socially desirable response bias or misreporting and learner purchasing behaviour may have been influenced during the tuckshop observation.

Comment 18

Lines 774-776: Make specific to children’s food environment, or indicate that analysis of children’s food environments is an important first step in characterizing food environments as a whole

Response 18

Thank you for your suggestion. We agree that describing children’s food environments is important but we suggest it is not necessarily the first step in understanding food environments. As noted in our conclusion, we suggested a broader understanding will help to improve public heath nutrition and the burden of malnutrition across all ages and this is key as most food environments overlap and intersect.

Reviewer 2 Report

Manuscript Number: nutrients-1227666

Title: The food environment of primary school learners in a low-to-middle-income area in Cape Town, South Africa

Summary 

Overall, this paper is well-written and fills an important gap in the literature. It outlines key directions for future research as it relates to the food environment in South Africa. The paper utilizes a multimethod approach and aims to conduct a case study to compile a comprehensive profile of the home, community and school food environment of primary school aged learners in order to advance the field of food environment research in the South African context. The paper can be improved by adding additional clarity to the methods, analysis and results.

Specific Comments/Questions: 

Abstract 

Line 28-29-Typo. Please add the word ‘and’ right before ‘healthfulness of foods…’

Methods

2.2

Line 95: Can you further define what a spaza shop is?

2.3.2: Was any software used to code the data? If data was hand-coded, please specify that. Also, can you report on inter rater reliability between the two coders such as Cohen’s kappa?

2.4: What data collection instrument was used in the tuck shop observation? For example, was it modified from a validated or existing tool e.g., NEMs?

2.5.4: Can you provide more detail on coding list development e.g., was it inductive or deductive? It seems that codes were created a priori and then coded quantitatively.

Can you provide a reference for this specific method of coding and any theoretical framework that informed methods and analysis?

Line 196: Can you provide a measure of inter rater agreement here as well?

Please include interview guides for the students and principal as well as codebooks as supplemental material.

2.6.1

Line 206: Can you define household representative criteria a bit more, was this the same as the primary caregiver or could it be different like a sibling?

2.6.2

Why were only four questions from the questionnaire used?

2.6.3

Can you provide any validation or reliability information on these surveys?

Results

3.1

Do these three settings correspond to themes? I would expect inductive thematic analysis to produce different groupings, as these seem a priori.

3.1.2

For these results, can you add in quotations to the text or summarize themes, subthemes and example quotes in a table?

3.4.2

Lines 477-478: Typo. Please add the word ‘and’ before ‘liked tasting new vegetables.’

3.4.3

Lines 493-494: Here, General Dealers are defined separately from spaza shops whereas they seem like the same thing on line 95. Please clarify

Discussion

Line 630: This concept of limited access could benefit from a brief add a discussion/definition of food deserts here.

Author Response

The food environment of primary school learners in a low-to-middle-income area in Cape Town, South Africa

We would like to thank the reviewers for their time and attention spent reviewing and for their helpful commentary.

Reviewer 2

Abstract 

Comment 1

Line 28-29-Typo. Please add the word ‘and’ right before ‘healthfulness of foods…

Response 1

Please see edit on page 1 lines 28-29

Comment 2

Line 95: Can you further define what a spaza shop is?

Response 2

Page 3 lines 171-176 now reads:

Informal food outlets include General Dealers (non-registered retail stores) ‘Spaza’ shops (the most common food outlet within the informal sector, often located in townships and poorer neighbourhoods), street vendors (cheap, often lower quality foods sold from roadside stalls, may be permanent or highly mobile) and home shop vendors (inexpensive, poor quality foods sold from a small shop in the front of a home) [31].

Comment 3

2.3.2: Was any software used to code the data? If data was hand-coded, please specify that. Also, can you report on inter rater reliability between the two coders such as Cohen’s kappa?

Response 3

Thank you for your comment. Page 5 lines 335-346 now reads:

Interviews were transcribed verbatim by NP. Each interview was read independently by MS and NP to identify common themes across the interviews. A coding list was developed inductively by hand in interactive sessions between MS and NP. Each theme was assigned a 4-digit code: the first, second, third and fourth digits represented the theme, subtheme, sub-subtheme and the sub-sub-subtheme respectively [34]. For each interview, any reference to a theme was coded accordingly and the frequency of mention for each code was recorded in an Excel spreadsheet and summed to give a total for each theme by NP (see coding framework in Supplementary materials). For quality control purposes MS and a third researcher (SO’H) (who read the interviews independently), then checked the code allocation interactively. Any discrepancies were discussed and a consensus agreed on between MS and SO’H.Themes and subthemes are mentioned specifically in the results, with sub-sub and sub-sub-subthemes reflected in examples given and text quotes.

As we understand it, the calculation of the Cohen’s Kappa to check inter-rater agreement  would not be appropriate  as checking of coding of interviews was done interactively and not independently. This has now been made clear in the revised manuscript as indicated above

Comment 4

2.4: What data collection instrument was used in the tuck shop observation? For example, was it modified from a validated or existing tool e.g., NEMs?

Response 4

Our data collection instrument used in the tuckshop observation was based on the tuckshop inventory conducted. At the time there was no validated tool that was appropriate for our study that we could adapt to suit our setting.

Please refer to page 3 lines 215-272.

Comment 5

2.5.4: Can you provide more detail on coding list development e.g., was it inductive or deductive? It seems that codes were created a priori and then coded quantitatively.

Response 5(please also refer to our response to comment 3)

Thank you for your comment. Codes were not identified priori; the analysis was inductive where the interviews were read and photographs viewed, emerging themes and subthemes were noted, considered, rearranged and reconsidered until a coding list had been generated. The interviews were then coded using the code list and the mentions summarised in tables, which is standard practice in this type of analysis. The frequency mentions are not used to identify most important ‘causes’, as the methodology is not quantitative, so any mention, even if by only one, is considered to be important.

Qualitative data analysis tends to be inductive, which means that the researcher identifies categories in the data, without predefined hypotheses.

Comment 6

Can you provide a reference for this specific method of coding and any theoretical framework that informed methods and analysis?

Response 6

We have used the reference below as a code for our coding. Please refer to page 3 line 209.

Braun. V,; Clarke, V. Using thematic analysis in psychology. Qual Res Psychol. 2006;1;3(2):77-101.

Comment 7

Line 196: Can you provide a measure of inter-rater agreement here as well?

Response 7

Please note that the thematic analysis was executed interactively and not independently by MS and SO’H so the calculation of inter-rater agreement would not be appropriate. The text has been revised on page 3 lines 208-211:

The transcription was read independently (MS and SO’H) for familiarisation with the content. The interview data was thematically analysed interactively by MS and SO’H, where themes were identified and organised via an inductive process without analytical preconceptions [34]’

Comment 8

Please include interview guides for the students and principal as well as codebooks as supplemental material.

Response 8

Regarding the interview guides used for learner interviews, please note the following information in the methods section on p 4 lines 311-319

The semi-structured interviews were an adapted version of the SHOWED methodology. The SHOWED method involves structuring Photovoice interviews by posing a series of questions about the participant’s photos (in this study, 3-5 questions) and were framed to elicit a descriptive response e.g. “What kind of meal is this?” “How often do you eat these types of foods?” “Where do you or your family buy this?” Additionally and guided by NP, learners discussed how their photos related to their diet and food availability and accessibility in their environment [44].

The principal interview guide has now been added as supplementary material as suggested.

Comment9

Line 206: Can you define household representative criteria a bit more, was this the same as the primary caregiver or could it be different like a sibling?

Response 9

Page 5 lines 355-357 now reads:

Learners were also invited to engage a representative from their households (e.g. a mother, father, grandmother, aunt, sister or brother) to complete a household questionnaire.

Comment 10

Why were only four questions from the questionnaire used?

Response 10

Only those questions that reflected environmental influences were included. The remainder will be published in a further paper.

Page 5 lines 353-355now reads:

Four questions from this questionnaire which were most relevant to this study are reported on in this paper.

Comment 11

Can you provide any validation or reliability information on these surveys?

Response 11

The questionnaire development process involved the following steps: firstly, the development of a conceptual framework of home food environmental influences based on the literature was conducted by an expert group consisting of two senior nutrition researchers and four postgraduate students in dietetics (construct and content validity).

Then the identification, consideration and adoption of items from published food environment research questionnaires/instruments (reference in the manuscript) and generation of new questions, assessment of the items in the draft questionnaire for appropriate coverage of core concepts, language level (readability and appropriateness of question formulation) was performed by two senior researchers.

Another independent researcher who was not involved in the study, but who had unique insight and extensive experience within the population (content and face validity) also contributed to these tasks. Reliability was ensured by extensive training of fieldworkers who conducted interviews with the learners and instructions to learners to assist parents with completion of questionnaires as necessary.

Comment 12

Do these three settings correspond to themes? I would expect inductive thematic analysis to produce different groupings, as these seem a priori.

Response 12

Thank you for your comment. Although we were interested in the different food environments, we did not set a priori. Instead, they emerged from the data.

Page 6 lines 451-452 now reads:

The three environmental settings, namely school, community and home environmental settings emerged as themes, and subtheme results are presented accordingly.

Comment 13

For these results, can you add in quotations to the text or summarize themes, subthemes and example quotes in a table?

Response 13

The principal consented to be interviewed and for a summary of his views to be included in the results, but did not consent to be quoted verbatim.

Page 3 lines 212-213

Verbatim quotes that illustrate themes/subthemes were not retrieved as the principal consented only to publication of an approved summary of his views.

Comment 14

Lines 477-478: Typo. Please add the word ‘and’ before ‘liked tasting new vegetables.’

Response 14

Page 13 lines 685-686 now reads:

The majority of respondents agreed they like the taste of most fruit, liked most vegetables, liked tasting new fruit and liked tasting new vegetables (Table 2).

Comment 15

Lines 493-494: Here, General Dealers are defined separately from spaza shops whereas they seem like the same thing on line 95. Please clarify

Response 15

Page 3lines 171-174 now reads:

Informal food outlets include General Dealers (non-registered retail stores) ‘Spaza’ shops ( the most common food outlet within the informal sector, often located in townships and poorer neighbourhoods) ,

Comment 16

This concept of limited access could benefit from a brief add a discussion/definition of food deserts here.

Response 16

Page 21 lines 924 – 926 now reads:

Limited access and availability, (commonly described as ‘food deserts’ where healthy foods such as F&V are insufficient [74]) can therefore mean less shopping trips.

Round 2

Reviewer 2 Report

Thank you for making the suggested changes. I am satisfied with the edits made! I could did not see the supplemental file of the interview guide so as long as that is included in the final publication I am fine with it.